

# Measurement phase transitions in the no-click limit as quantum phase transitions of a non-hermitean vacuum

Caterina Zerba[1,2,3,4] and Alessandro Silva[1⋆]

**1** International School for Advanced Studies (SISSA), via Bonomea 265, 34136 Trieste, Italy
**2** Universitá degli Studi di Trieste, via Alfonso Valerio 2, 34127 Trieste, Italy
**3** Department of Physics, Technical University of Munich, 85748 Garching, Germany
**4** Munich Center for Quantum Science and Technology (MCQST),
Schellingstr. 4, 80799 München, Germany

⋆ asilva@sissa.it

## Abstract

We study dynamical phase transitions occurring in the stationary state of the dynamics of integrable many-body non-hermitian Hamiltonians, which can be either realized as a no-click limit of a stochastic Schrödinger equation or using spacetime duality of quantum circuits. In two specific models, the Transverse Field Ising Chain and the Long Range Kitaev Chain, we observe that the entanglement phase transitions occurring in the stationary state have the same nature as that occurring in the vacuum of the non-hermitian Hamiltonian: bounded entanglement entropy when the imaginary part of the quasi-particle spectrum is gapped and a logarithmic growth for gapless imaginary spectrum. This observation suggests the possibility to generalize the area-law theorem to non-Hermitian Hamiltonians.

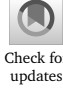

# 1 Introduction

Entanglement is one of the key properties [1] characterizing the statics and dynamics of quantum many-body systems [2,3]. The scaling of the entanglement entropy of a subsystem with respect to its linear size $L$, for example, gives clear indications on its simulability, e.g. with tensor network algorithms [4]. For ground states, entanglement is closely connected to spectral properties [5–7]: in one dimension for example depending on whether the system is gapped or gapless one can expect either a bounded entanglement entropy or logarithmic scaling with subsystem size [4,8]. With the notable exception of quantum scars [9], for highly excited states the entanglement entropy is instead expected to obey the volume law [10–15] as the thermodynamic one does. For quantum chaotic systems, this is direct consequence of the eigenstate thermalization hypothesis [16] which implies that coherent thermal states cannot be distinguished from mixed ones on the basis of bipartite entanglement. Only a more sophisticated look at entanglement, for example focusing on its multipartiteness [17,18] reveals the nontrivial entanglement structure behind coherent thermal states.

Entanglement is either produced via interactions or degraded by dephasing and measurements [19]. As thermodynamic phase transitions result from the competition between minimization of energy and maximization of entropy, the competition between interaction induced unitary dynamics and measurements leads to phase transitions witnessed by entanglement [20–22]. Unitary dynamics in both circuit models [23–32] and many-body systems [33–36] when punctuated by either projective or weak measurements, gives rise to an entanglement phase transition where the two phases correspond to an area law one (strong mesurement phase) and volume law (weak measurement). While circuit models allow a simple understanding of these transitions in connection to models in classical statistical mechanics [37,38], a comparative phenomenology is observed in the unitary dynamics of interacting systems subject to continuous measurements [39] (as described by a stochastic Schrödinger equation [19]). While this limit, which is in a sense more realistic [40–43], is only amenable to numerical simulations, the dynamics of the quantum trajectory corresponding to the no-click limit is reduced to that generated by a non-hermitean Hamiltonian [44,45]. Interestingly, this trajectory, even though exponentially improbable, displays the same transition observed in the presence of quantum jumps for these models. Understanding the origin and generality of this fact could simplify the analysis of these transitions in various cases, for example in the case of long-range interactions [38,46–48].

In this paper we give a contribution in this direction by revisiting Measurement-Induced phase transitions in the zero-click trajectories of the quantum Ising chain subject to local transverse field measurements. In addition we analyze the zero-click dynamics of a long-range Kitaev Model in one dimension subject to density measurements. We show that for both models, which are integrable in terms of free fermions, the entanglement properties of the vacuum of the non-Hermitian Bogoliubov quasiparticles are the same of the steady state of the zero-click trajectories. In addition, we observe that the entanglement properties of the vacuum can be understood as a generalization of the area-law theorem for the ground-state of Hermitian Hamiltonians [4,8]: with a gap in the *imaginary* part of the spectrum the bipartite entanglement entropy is bounded while when the gap closes the entanglement entropy scales logarithmically with the dimension of the subsystem. More specifically, for the Long-range Kitaev Model, where the long range pairing decays with distance as a power-law with exponent $d$ two regimes are found: in the first, $d < 1$, a phase transition occurs from a logarithmic phase to another logarithmic phase at a critical measurement rate $\gamma = \gamma_c$. Even for $\gamma > \gamma_c$ the entanglement does not appear to be bounded: since if $d < 1$ the gap in the imaginary part of the spectrum is always closed this is consistent with the generalized area-law conjecture. For $d > 1$, we observe instead in the thermodynamic limit a phase transition from a phase where

the entanglement entropy obeys a logarithmic law ($\gamma < \gamma_c$) to an area-law phase. Since in this regime the gap opens if $\gamma > \gamma_c$ this is again consistent with the generalized area law theorem.

The article is organized as follows: in Section 2 the non Hermitian Hamiltonians for the two models are analyzed in detail; in Section 3 the results for the Transverse Field Ising Model are presented, and the area-law conjecture is introduced; in Section 4 the results for the Long Range Kitaev Chain are reported.

## 2 Models and zero-click limit

The main purpose of this work will be to study the dynamics of specific trajectories (i.e. sequences of measurement outcomes) in many-body systems coupled to a local measurement apparatus described in terms of positive operator-valued measurements (POVM) [19]. In general in this limit the stochasticity of the outcome of the measurements imply that the variation of the wavefunction of the system at each time step is random as well and described by a stochastic Schrödinger equation [19]. For example for a quantum Ising chain

$$H_0 = -J \sum_{i=1}^{N} \hat{\sigma}_i^x \hat{\sigma}_{i+1}^x - h \sum_i^N \hat{\sigma}_i^z, \tag{1}$$

subject to a continuous monitoring of on-site spin by the measuring operators $\hat{n}_i = \frac{1-\hat{\sigma}_i^z}{2}$ (and $\frac{1+\hat{\sigma}_i^z}{2}$), the equation takes the form [39]

$$d|\psi(t)\rangle = -i\hat{H}_0|\psi(t)\rangle\,dt - \frac{\gamma}{2}\sum_i\left(\langle\hat{n}_i\rangle - \hat{n}_i\right)|\psi(t)\rangle\,dt + \sum_i \delta N_i(t)\left(\frac{1-\hat{n}_i}{\sqrt{\langle 1-\hat{n}_i\rangle}} - 1\right)|\psi(t)\rangle. \tag{2}$$

Here $\left\{\delta N_i(t)\right\}_{i=1,..,N}$ are stochastic variables that describe the measurement outcomes ($\delta N_i(t) = 1$ when the measurement apparatus clicks, i.e. a local down projection is measured, zero otherwise). Notice that if $\delta N_i = 0$ for all sites and for a certain interval of time (no-click limit), then the evolution of the system is determined by the effective Hamiltonian $\hat{H} = \hat{H}_0 - i\frac{\gamma}{2}\sum_i\left(\langle\hat{n}_i\rangle - \hat{n}_i\right)$. If on the other hand $\delta N_i = 1$ for some $i$ then the wave-function changes discontinuously and the evolution has a quantum jump. The set of values of the variables $\delta N_i(t)$ determines the realization of the interaction of the external environment with the system.

In the no-click limit the effective Hamiltonian determining up to a c-number the evolution of the wave function is

$$H_{\mathrm{QI}} = -J \sum_{i=1}^{N} \hat{\sigma}_i^x \hat{\sigma}_{i+1}^x - \left(h + i\frac{\gamma}{4}\right) \sum_i^N \hat{\sigma}_i^z, \tag{3}$$

and the evolution is given by

$$|\psi(t)\rangle = \frac{e^{-i\hat{H}_{\mathrm{QI}}t}|\psi_0\rangle}{\|e^{-i\hat{H}_{\mathrm{QI}}t}\psi_0\|}. \tag{4}$$

A similar measurement setting can be thought for a fermionic long-range version of Eq.(3) where, after the Jordan-Wigner transformation, a long range pairing term is introduced. The long range pairing term connects sites at all distances with a coupling constant decaying as a power law (the Long-Range Kitaev Chain in D=1 spatial dimension)

$$H_{\mathrm{K}} = -J \sum_i (\hat{c}_i^\dagger \hat{c}_{i+1} + h.c.) - \frac{J}{2}\sum_i \sum_{r=1}^{L-1} \frac{1}{l^d}\left[\hat{c}_i^\dagger \hat{c}_{i+r}^\dagger + h.c.\right] - \left(h + i\frac{\gamma}{4}\right)\sum_i(1 - 2\hat{c}_i^\dagger \hat{c}_i), \tag{5}$$

where the power $d$ controls the decay as a function of the distance $l$ (that is $l = \min\{r, L - r\}$, we use antiperiodic boundary conditions). The procedure used for the diagonalization of the quantum Ising chain can be applied here as well.

The advantage of the no-click limit is that in this case the Hamiltonian can be mapped exactly in a free-fermionic model with nearest neighbour hopping by means of the Jordan-Wigner transformation; one then proceeds diagonalizing the Hamiltonian by means of a generalized Bogoliubov rotation. All the details concerning the diagonalization can be found in Appendix A. For both models the diagonalized Hamiltonian takes the form

$$\hat{H} = \sum_{k>0} \lambda_k \hat{\gamma}_k^* \hat{\gamma}_k - \lambda_k \hat{\gamma}_{-k} \hat{\gamma}_{-k}^* = \sum_{k>0} \lambda_k (\hat{\gamma}_k^* \hat{\gamma}_k + \hat{\gamma}_{-k}^* \hat{\gamma}_{-k}) - \Lambda_0 \,, \tag{6}$$

where $\Lambda_0 = \sum_{k>0} \lambda_k$, $\hat{\gamma}$ are the non-hermitian quasiparticle annihilation operators and $\lambda_k$ are the ( complex ) eigenvalues which are specific of the model considered. We find that for the Quantum Ising model

$$\lambda_k = \pm \sqrt{4\left(h - J \cos k + i\frac{\gamma}{4}\right)^2 + 4J^2 \sin^2 k} \,, \tag{7}$$

while for the Long Range Kitaev model

$$\lambda_k = \pm \sqrt{4\left(h - J \cos k + i\frac{\gamma}{4}\right)^2 + J^2 g_d(k)^2} \,, \tag{8}$$

where $g_d(k) = \sum_{r=1}^{L-1} \frac{\sin(kr)}{l^d}$. The sign is chosen in such a way that the sign of the imaginary part of $\lambda_k$ is negative [49]. Thus the non-hermitian Hamiltonian right vacuum

$$\hat{\gamma}_k |0_\gamma\rangle = 0 \,, \qquad \hat{\gamma}_{-k} |0_\gamma\rangle = 0 \tag{9}$$

can always be construct as the state with largest imaginary part (not lowest real part). Because of the normalization factor in Eq.(4), the stationary states of the dynamics will be a linear combination of the vacuum and of quasi-particle states such that $\Gamma_k \equiv \text{Im}[\lambda_k] = 0$, while all amplitudes of the other quasi-particle states will decay to zero at long times.
Armed with full knowledge of the quasi-particles describing the zero-click limit we are now ready to describe the stationary state of the non-hermitean dynamics and its entanglement properties.

## 3   Quantum Ising chain and generalized area law conjecture

Let us now focus on the dynamics of the system: in order to be specific we will consider as initial state

$$|\psi_0\rangle = \otimes_i^L |0_c^i\rangle \,, \tag{10}$$

where $|0_c^i\rangle$ is defined as the vacuum of the Jordan-Wigner fermions on the site $i$. Notice however that most of the arguments presented below are not dependent on this choice. This state can be written as a linear combination of all the eigenstates of the non-Hermitian Hamiltonian. After a long time of evolution only the unsuppressed states of the initial linear combination, the vacuum and eventually the state with the two quasiparticles $\hat{\gamma}_{-q^*}^*$ and $\hat{\gamma}_{q^*}^*$, will give a non negligible contribution to the linear combination, i.e.

$$|\psi(t)\rangle = \frac{e^{-i\hat{H}t} |\psi_0\rangle}{||e^{-i\hat{H}t} |\psi_0\rangle||} \approx A |0_\gamma\rangle + e^{i\phi(t)} B |q^*, -q^*\rangle \,, \tag{11}$$

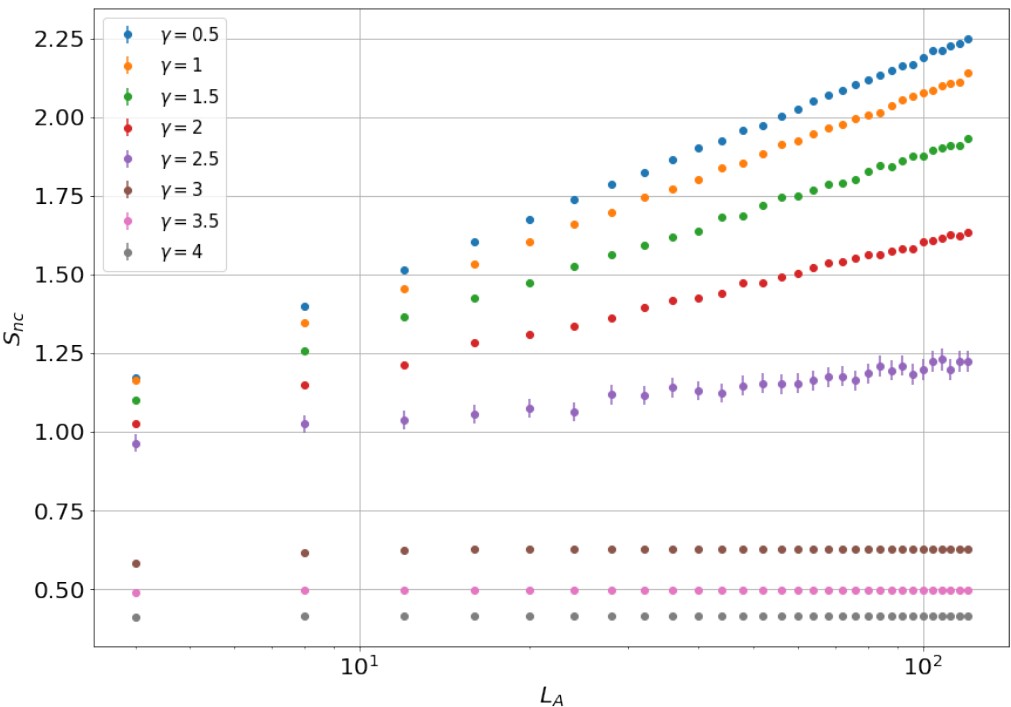

Figure 1: Scaling of the average entanglement entropy as a function of the dimension of the interval $L_A = \frac{L}{4}$, $h = \frac{1}{\sqrt{2}}$. The oscillations are due to the relative phase between $|0\rangle$ and $\hat{\gamma}^*_{-q^*}\hat{\gamma}^*_{q^*}|0\rangle$ and have been dealt with by averaging over the time, ideally $S_{nc} = \lim_{T\to\infty}\frac{1}{T-T_0}\int_{T_0}^{T}dt\,S(t)$, in accordance with the method used in [39].

where $|q^*,-q^*\rangle = \hat{\gamma}^*_{-q^*}\hat{\gamma}^*_{q^*}|0_\gamma\rangle$, $q^*$ defined as the momentum in the First Brillouin Zone where $\Gamma_{q^*} = 0$ if such momentum exists, and $\phi(t)$ is the relative phase between the two kets due to the evolution. The same argument holds for all possible initial states, provided that they have a non-zero superposition with $|0_\gamma\rangle$ and $|q^*,-q^*\rangle$. The explicit expression of the steady state after a long time of evolution allows us to compute the entanglement entropy of an interval of size $L_A$ with respect to the rest of the system by means of the Majorana Fermions correlation matrix (see Appendix B). The result is shown in Fig. 1 for the quantum Ising chain, for various $\gamma$ as a function of system size. As previously discussed in Ref. [39, 50] it clearly displays a dynamical transition as $\gamma$ is increased towards a critical $\gamma_c(h) = 4\sqrt{1-h^2}$: for $\gamma < \gamma_c(h)$ the entropy scales logarithmically with the interval size $L_A$, while for $\gamma > \gamma_c(h)$ the entropy is constant and satisfies an area law. Interestingly, for $\gamma < \gamma_c(h)$ the quasi-particle spectrum has a gapless imaginary part at $q^* = \arccos h$ (corresponding to $\Gamma_{q^*} = 0$, see Fig.2) while for $\gamma > \gamma_c(h)$ a gap opens in the imaginary part of the spectrum [50]. Therefore while below $\gamma_c$ the stationary state is a linear superposition of $|0_\gamma\rangle$ and $|q^*,-q^*\rangle$ for $\gamma > \gamma_c$ this two particle state is absent.

Even though apparently the entanglement phase transition is associated to the disappearance of the state $|q^*,-q^*\rangle$ from the stationary state, we will show that it is actually a consequence of the properties of the non-hermitean vacuum $|0_\gamma\rangle$. As shown in Fig. 2, where the scaling of the entanglement entropy as a function of the size of the interval is reported , the entropy computed on the vacuum has exactly the same properties as the stationary state. It is however intriguing that, as in the case of ground state quantum phase transitions, where the dichotomy between bounded and logarithmic entanglement is associated to the presence or absence of a gap in the spectrum [4, 8], in this example an analogous conjecture could be formulated for the entanglement properties of the vacuum of non-hermitian hamiltonians: a bounded entan-

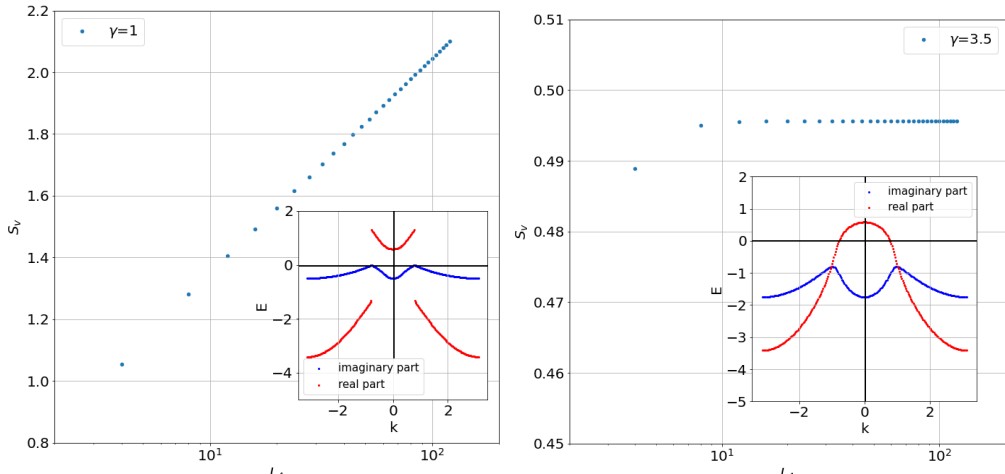

Figure 2: *Left*: Entanglement entropy of the vacuum as a function of the dimension of the interval $L_A = \frac{L}{4}$, $h = \frac{1}{\sqrt{2}}$; Inset: L=250, real and imaginary part of the hamiltonian eigenvalues as a function of k in the FBZ, lattice spacing a=1, $J = 1$, $h = \frac{1}{\sqrt{2}}$. *Right*: Entanglement entropy of the vacuum as a function of the dimension of the interval $L_A = \frac{L}{4}$, $h = \frac{1}{\sqrt{2}}$; Inset: L=250, real and imaginary part of the hamiltonian eigenvalues as a function of k in the FBZ, lattice spacing a=1, $J = 1$, $h = \frac{1}{\sqrt{2}}$.

glement entropy is associated to a gapped *imaginary* part of the spectrum while logarithmic growth is observed otherwise(see insets of Fig. 2).

In Fig 3 it is shown that the contribution of the two-particle state is inessential by plotting the relative difference between the entanglement entropy in the vacuum and the entanglement entropy in the steady state. It is not surprising that the entanglement properties are encoded in the non-hermitian vacuum. Indeed, although the vacuum does not contain non-Hermitian quasiparticles, it is a state rich in Hermitian quasiparticles $\hat{\eta}_k$, $\hat{\eta}_k^\dagger$ diagonalizing the quantum Ising chain for $\gamma = 0$. We can indeed write it as

$$\left|0_\gamma\right\rangle = \prod_{k>0}(\hat{\gamma}_k\hat{\gamma}_{-k})\left|0_\eta\right\rangle = \prod_{k>0}(\alpha_k + \beta_k\hat{\eta}_{-k}^\dagger\hat{\eta}_k^\dagger)\left|0_\eta\right\rangle, \tag{12}$$

where $\alpha_k$ and $\beta_k$ are normalized coefficients, that can be easily obtained by writing the quasiparticles $\hat{\gamma}_k$ in terms of the Jordan-Wigner fermions $\hat{c}_k$ and then substituting them with their expression in terms of the quasiparticles $\hat{\eta}_k$.

The logarithmic scaling is a remarkable result if we consider that for ground states such scaling is expected only for critical states. In the present case, the critical scaling of the entanglement entropy in the gapless phase (of the imaginary part of the spectrum) is expected to be $S(L_A) = c\ln(L_A) + b$ both for the one computed on the stationary state and in the no-click limit. In order to study the dependence of $c$ on the parameters $\gamma, h$, one can consider the incremental ratio of the bipartite entanglement entropy when a site is added to the considered interval, $\Delta S = S(L_A + 1) - S(L_A)$: clearly in an area law phase the variation is equal to zero, while in the logarithmic phases we have $\Delta \sim \frac{c}{L_A}$ in the thermodynamic limit. We can then estimate $c$ as $c = \Delta S\, L_A$. The resulting $c(\gamma)$ for different $h$ is shown in Fig. 4.

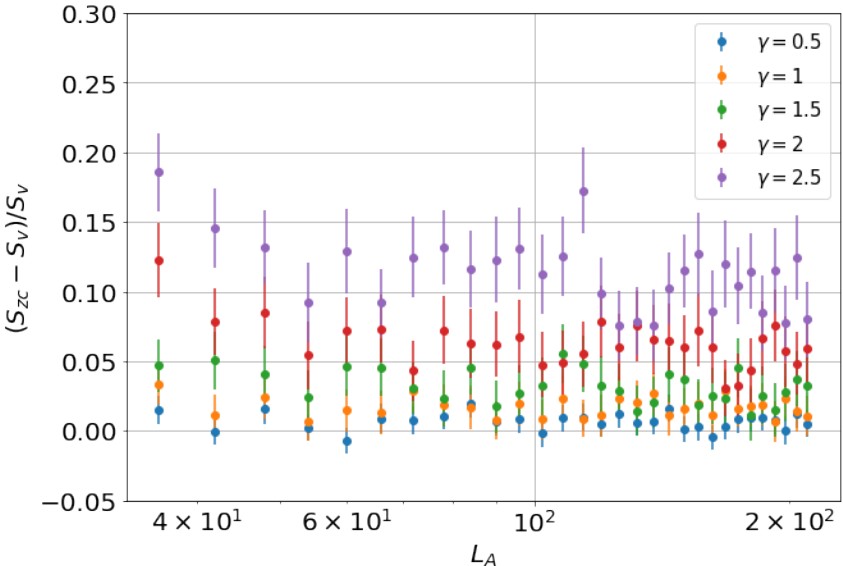

Figure 3: Difference between the entanglement entropy of the zero click trajectories and the one in the vacuum for different dimensions of the system $L$, and $L_A = \frac{L}{4}$ relative to the vacuum entanglement entropy.

## 3.1 Perturbed no-click limit for the quantum Ising chain

The phase transition observed in the zero-click limit appear to have the same properties of the one observed in the presence of quantum jumps [39], a result that has been discussed and understood using semi-classical considerations [51]. We are now interested in extending the generalized area law conjecture away from the zero-click limit by considering the addition of rare jumps. We have found that the properties of interest (the scaling of the entanglement entropy) are satisfied in this regime too. At the same time, we will see that the area law conjecture discussed in the previous section is satisfied in this limit as well.

In a typical trajectory the system at each time step has a certain probability to jump,

$$\delta p = \delta t \sum_l \gamma \langle 1 - \hat{c}_l^\dagger \hat{c}_l \rangle. \tag{13}$$

Each jump introduces in the system quasiparticles with all possible momenta. In the following we will consider trajectories where the jumps are very sparse. The perturbed zero-click limit in particular is defined as the limit in which the time between jumps is sufficiently long so as to make all quasiparticles with a finite lifetime decay. In this case right before every other jump we have that

$$|\psi(t)\rangle := A|0_\gamma\rangle + e^{i\phi(t)}B|q^*, -q^*\rangle, \tag{14}$$

if $\gamma < \gamma_c$ and where $\phi(t)$ is the relative phase between the two kets due to the real part of the eigenvalues of the Hamiltonian. The parameters $A$ and $B$ are the only parameters needed to describe the jump. If $\gamma > \gamma_c$ the steady state after long time is instead $\psi(t) = |0_\gamma\rangle$.

In the perturbed no-click limit, the study of the evolution of the system due to jumps is reduced to the study of the evolution of the two parameters A and B. Indeed while a jump generates quasi-particles of all momenta they will again all decay except the ones with momentum $\pm q^*$, i.e. right before the new jump the state will be again of the form $|\psi\rangle = A'|0_\gamma\rangle + B'|q^*, -q^*\rangle$. Hence the description of the evolution due to jumps is reduced to the characterization of the discrete map $(A, B) \rightarrow (A', B')$.

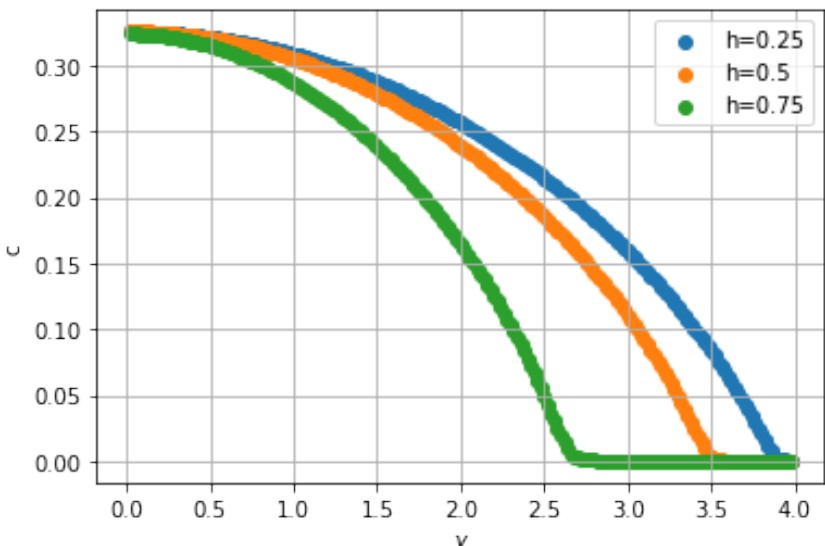

Figure 4: Value of the parameter c as a function of $\gamma$, where c is defined as $S = c\ln(L_A) + b$. Different values of the magnetic field are considered.

For $\gamma < \gamma_c$ starting from the state $|\psi\rangle = A|0_\gamma\rangle + B|q^*, -q^*\rangle$ the wave-function after a jump is

$$|\psi(t)\rangle \to |\psi(t+dt)\rangle = \frac{(1 - \hat{c}_n^\dagger \hat{c}_n)}{\sqrt{\langle 1 - \hat{n}_n \rangle_t}}|\psi(t)\rangle. \tag{15}$$

In order to compute $|\psi(t+dt)\rangle$ we have to write $\hat{c}_n^\dagger \hat{c}_n = 1/L \sum_{k,k'} e^{i(k'-k)R_n} \hat{c}_k^\dagger \hat{c}_{k'}$ in terms of the non-hermitean quasi-particle modes $\hat{\gamma}_k$.

The expressions depend on the sign of $k, k'$: there are four cases $k, k' > 0$; $k < 0, k' > 0$; $k > 0, k' < 0$; $k, k' < 0$. For example for $k > 0, k' > 0$,

$$\sum_{k,k'>0} e^{i(k-k')R_n} \hat{c}_{k'}^\dagger \hat{c}_k = \sum_{k>0,k'>0} e^{i(k-k')R_n} \frac{1}{\det(V_{k'})}(u_{k'}\hat{\gamma}_{k'}^* - v_{k'}\hat{\gamma}_{-k'})(u_k\hat{\gamma}_k + v_k\hat{\gamma}_{-k}^*),$$

where

$$V_k = \begin{pmatrix} u & -\frac{\lambda_k - a}{b^*}u^* \\ \frac{\lambda_k - a}{b}u & u^* \end{pmatrix}, \tag{16}$$

and $u_k = 1/\sqrt{1 + |\frac{\lambda_k - a}{b}|^2}$, $v_k = \frac{\lambda_k - a}{b}u$, $a = 2(h - J\cos(k)) + i\frac{\gamma}{2}$, $b = 2iJ\sin(k)$.

Applying this operator as well as those with different combinations of $k, k'$ to the state Eq. 14 and letting all states with $k \neq q^*$ decay according to the assumptions of this limit we finally obtain that if $t' \gg t + dt$ and no other jumps occurred in the interval [t,t'] the wave function is

$$|\psi(t')\rangle \approx \left[A - \frac{2}{L}\left(\sum_{k>0}|u|^2\frac{(\lambda_k - a)^2}{|b|^2}A + \frac{1}{\det(V_q)}\frac{\lambda_q - a}{b}u^2 B\right)\right]|0_\gamma\rangle \tag{17}$$

$$+ \left[B - \frac{2}{L}\left(\frac{1}{\det(V_q)}u^{*2}\frac{\lambda_q - a}{b^*}A + \left(\frac{|u|^2}{\det(V_q)} + \sum_{k>0,k\neq q}\frac{|u|^2}{\det(V_k)}\frac{(\lambda_k - a)^2}{|b|^2}\right)B\right)\right]|q^*, -q^*\rangle.$$

Since at each step the state that describes the system is normalized and independent of a global phase, the complex ratio $x = \frac{A}{B}$ is the only relevant quantity; hence only two degrees of freedom describe the dynamics of the system. Furthermore, the non Hermitian evolution

provides a stochastic phase between $A$ and $B$ due to the different (real part of the) energy between $|0_\gamma\rangle$ and $|q^*, -q^*\rangle$. At each step of the dynamics we therefore have $x \to e^{i\phi(t)}x'$, with

$$x' = \frac{(C_1 - 1)x + C_2}{C_3\, x + C_4 + C_5 - 1}, \qquad (18)$$

where

$$C_1 = \sum_k \frac{2}{L \det(V_k)}\left(\frac{(\lambda_k - a)^2}{|b|^2}|u|^2\right), \qquad C_2 = \frac{2u^2}{L \det(V_q)}\frac{\lambda_q - a}{b}, \qquad C_3 = \frac{2u^{*2}}{L \det(V_q)}\frac{\lambda_q - a}{b^*},$$

$$C_4 = \frac{2|u|^2}{L \det(V_q)}, \qquad C_5 = \sum_{k>0, k \neq q} \frac{2|u|^2}{L \det(V_k)}\frac{(\lambda_k - a)^2}{|b|^2}.$$

The stochasticity of the time between jumps is encoded in the stochasticity of the relative phase $e^{i\phi(t)}$.

In particular the time between one jump and the following is $\Delta t = \lambda \delta t + \tau$, where $\lambda$ is the number of steps necessary to let all modes with $k \neq q^*$ decays and $\tau$ is generated randomly by assuming it is distributed according to an exponential distribution with average time $\tau = \frac{1}{\delta p}$, $\delta p = \gamma \delta t \left\langle \sum_i (1 - c_i^\dagger c_i)\right\rangle_\psi$.

In order to better understand what are the key ingredients of the dynamics let us first consider the map in the absence of a stochastic phase, $x \to x'$. Under this assumption the map has two fixed points and after few iterations it collapses on the attractor. In the corresponding state the parameter $c$ characterizing the log-growth of the entanglement entropy as a function of the dimension of the subsystem is reported in figure Fig.5(a). The apparent jump observed at the transition turns out to be an artifact of setting the phase $\phi = 0$. Indeed, if the jumps are performed periodically with a fixed phase $x \to e^{i\phi}x', e^{i\phi} \neq 1$, the position of the fixed points changes and at the transition the incremental ratio is smooth (see Fig.5(b)).

Let us now consider the stochastic jump dynamics in this limit averaging the entropy over several trajectories at time $T = N_o \lambda \delta_t + \sum_{i=1}^{N_o} \tau_i$, where $N_o$ is the number of jumps and $\tau_i$ are intervals of time stochastically generated for a system that has not jumped in the previous $\lambda \delta t$ time. In the perturbed zero-click trajectories the scaling of the entanglement entropy witnesses a transition in $\gamma = \gamma_c$ from a logarithmic to a bounded entanglement phase. In both 5(b) and 5(c) the coefficient $c$ of the logarithm, $S_1 = c \ln(L) + b$, seems to go to zero without discontinuity for large values of $\gamma$: at $\gamma \sim \gamma_c$ a smooth cross-over instead of a phase transition can be found, a fact that we interpret as a finite-size effect, as in [39]; the transition point is however unchanged by the presence of rare jumps. We have seen that the perturbed zero-click limit shows clearly an entanglement phase transition as for the zero-click limit. In particular, since the form of the state is always as in Eq.(14), the entanglement will be essentially determined, except for a transient, by the quasi-particle vacuum. Hence the validity of the perturbed zero-click limit implies that of the generalized area-law conjecture.

The problem however of putting the connection above on solid mathematical grounds remains open because the perturbed zero-click limit is an approximation valid only for finite size systems and away from the critical point $\gamma_c$.

In order to see this notice that we can estimate the condition of validity of the perturbed no-click limit by asking for instance that the suppression due to the deterministic evolution of

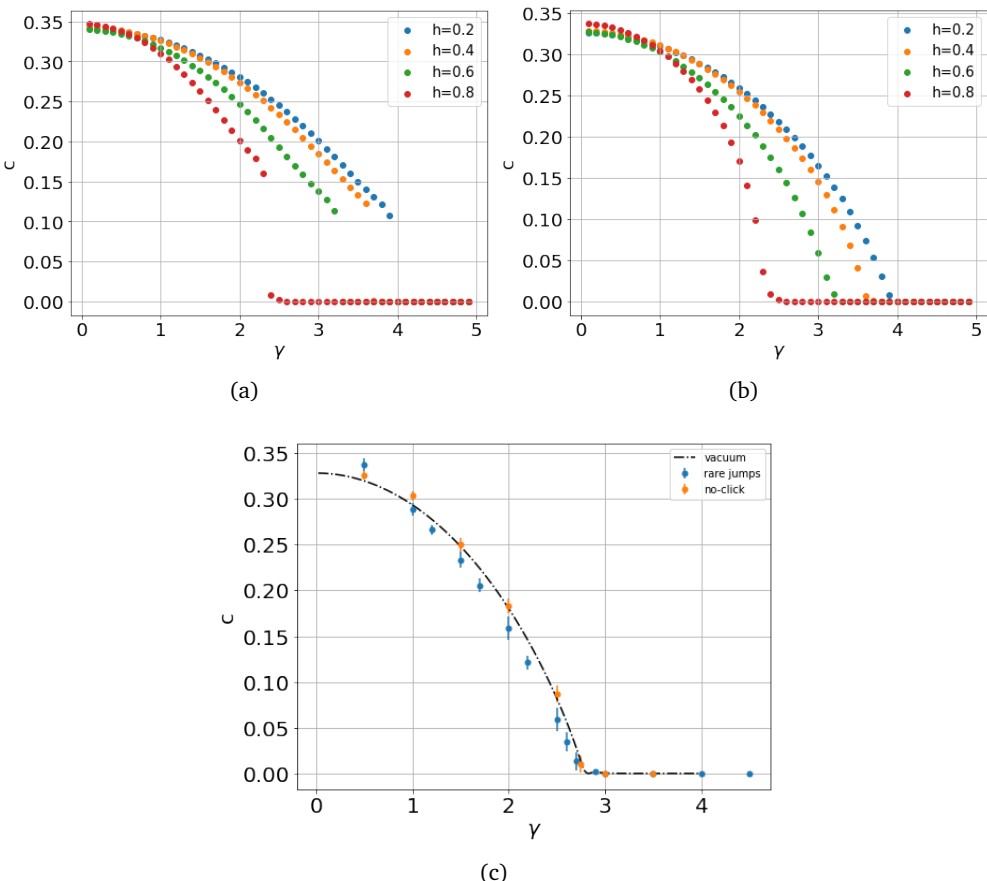

Figure 5: Variation of the parameter $c$, where $S_1 = c \ln(L_A) + b$,(a) in the attractor if $\phi(t) = 0$ as a function of the coupling with the bath $\gamma$, for different values of the magnetic field $L_A = \frac{L}{4}$, (b) in the attractor if $\phi(t) = \frac{\pi}{2}$ as a function of the coupling with the bath $\gamma$, for different values of the magnetic field,(c) obtained by fitting the scaling of the average entanglement entropy over $N_{tr} = 20$ trajectories for $h = \frac{1}{\sqrt{2}}$.

all decaying modes is at least of one order of magnitude. We can estimate $\lambda$ as

$$
\begin{aligned}
e^{-\lambda |\Gamma_k| \delta t} &\sim \frac{1}{10L} \\
\to \lambda &\sim \frac{\log(10L)}{|\Gamma_k| \delta t} .
\end{aligned}
\tag{19}
$$

Provided $\frac{\gamma^2}{\gamma_c^2} < 1 - \frac{1}{L}$ and for $k \simeq q^*$ we have

$$
\Gamma_k = \frac{\partial}{\partial k} \Gamma_k \bigg|_{q^*} (k - q^*), \qquad \frac{\partial}{\partial k} \Gamma_k \bigg|_{q^*} = \frac{\pm 2\gamma}{\sqrt{1 - \frac{\gamma^2}{\gamma_c^2}}} .
\tag{20}
$$

Inserting this expression in equation 19

$$
\lambda \delta t \sim \frac{L \log(10L) \sqrt{1 - \frac{\gamma^2}{\gamma_c^2}}}{2\pi \gamma} .
\tag{21}
$$

This quantity has to be compared with the the average time between one jump and the other,

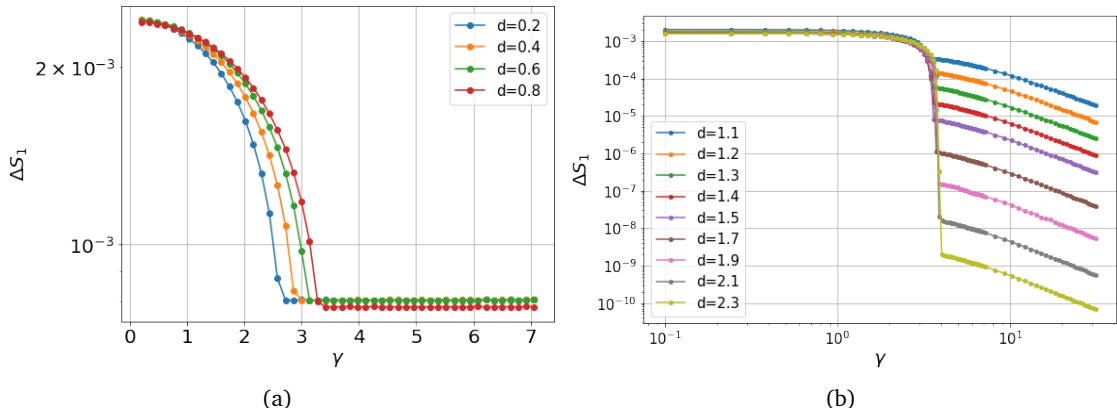

Figure 6: Incremental ratio of the Entanglement entropy, for $L_A = 200$, as a function of $\gamma$ for different values of $d$ in a system characterized by L=2000, h=0.1.

defined by the unit time probability of having a measure on each site

$$\delta_p = \delta_t \gamma \sum_i \left\langle c_i c_i^\dagger \right\rangle_\psi \sim \gamma \delta t \mathcal{M}, \tag{22}$$

where $\mathcal{M} = \sum_i \left\langle c_i c_i^\dagger \right\rangle_\psi$ is an extensive quantity. Assuming an exponential distribution [51], we obtain $\tau = 1/(\gamma \mathcal{M}) \propto 1/L$. It is evident from these estimates therefore that the decay of all modes except $q^*$ is complete if $\lambda \delta t < \tau$ which however can be satisfied only for sufficiently small systems, certainly not in the thermodynamic limit. In Eq.21 one could be tempted to argue that setting $\gamma \approx \gamma_c$ could help, however this is an artifact of the linear expansion used in Eq.20 and since close to $\gamma_c$ we still have $\Gamma_k \sim |k - q^*|^{\frac{1}{2}}$ that vanishes at $q^*$ it is easy to see that also in this case we are limited to small system sizes.

## 4 The long range Kitaev chain

Let us now turn to the no-click limit of our second model, the long-range Kitaev chain. In this section we will analyze the entanglement phase transition observed in the quasi-particle vacuum of this model. Notice however that the same conclusions can be reached by studying also in this case the stationary state of the no-click limit.

The long-range Kitaev model has two types of measurement induced phase transition depending on the value of the exponent that characterizes the power-law decaying interactions $\sim 1/r^d$.

Let us start by considering the regime $d < 1$. As discussed before, a compact way to detect the phase transition is to plot the incremental ratio of the entanglement entropy with respect to the subsystem size $\Delta S = S(L_A + 1) - S(L_A)$. As we see in Fig. 6(a) for $d < 1$ we clearly detect a phase transition as a discontinuity in the derivative of $\Delta S(\gamma)$. For $\gamma > \gamma_c$ the incremental ratio does not appear to vary significantly, in particular as the system size $L$ increases towards the thermodynamic limit. Most importantly, however, it is not zero but finite. In addition assuming that $\Delta S = c/L_A$ and therefore plotting $L_A \Delta S$ (see Fig. 7 (a)) we see that in both regime the entropy is clearly logarithmic and what has a singular behavior is the coefficient $c(\gamma)$ at the phase transition. Interestingly, the imaginary part of the quasi-particle spectrum is in both cases gapless, consistently with the generalized area-law conjecture. Indeed since

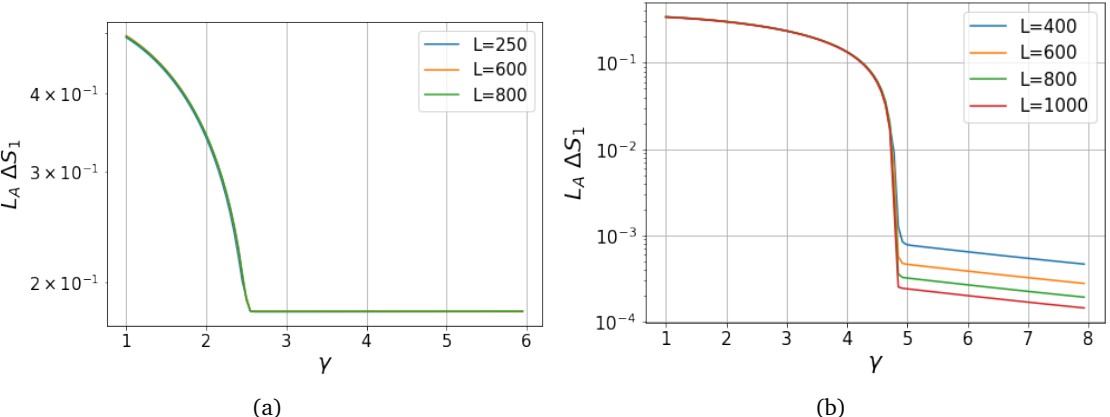

Figure 7: Incremental ratio of the entanglement entropy normalized by the length of the interval, in the vacuum, for h= 0.1, $L_A = \frac{L}{4}$, (a) d=0.2, (b) $d = 1.7$.

$\lambda_k = \pm \sqrt{4\left(h - J\cos k + i\frac{\gamma}{4}\right)^2 + J^2 g_d(k)^2}$ as reported in [8], the eigenvalues have zero imaginary part at finite momentum $q^*$, $\Gamma_{q^*} = 0$, if the relation $h = J\cos(q^*)$ is satisfied provided $\gamma < \gamma_c$ where $\gamma_c = 2Jg_d(q^*)$ (the function $g_d(q)$ in the thermodynamic limit $L \to \infty$ is known in literature as Clausen functions). However for $d < 1$ and $k \to 0$, at leading order

$$\lambda_k \approx \left|\cos\frac{\pi d}{2}\frac{\Gamma(1-d)}{k^{1-d}}\right|\left[1 + \left(\zeta(d-1)\cos\frac{\pi d}{2}\Gamma(1-d)k^{2-d}\right.\right.$$
$$\left.\left. + \left((h-J)^2 - \frac{\gamma^2}{16} + i\frac{\gamma}{2}(h-J)\right)\frac{k^{2d-2}}{\left(\cos\left(\frac{\pi d}{2}\right)\Gamma(1-d)\right)^2}\right)\right] + O(k^{3-d}).$$

Therefore the real part of the eigenvalue diverges as $\text{Re}\{(\lambda_k)\} \sim \frac{1}{k^{1-d}}$ while the imaginary part goes to zero as $\text{Im}\{(\lambda_k)\} \sim k^{1-d}$ for every value of $\gamma$. Because of that in the regime $d < 1$ in the thermodynamic limit the gap never opens in the imaginary part of the spectrum .

A completely different behavior is observed for $d > 1$ (see Fig.6(b) and Fig.7(b)): in this case while for $\gamma < \gamma_c$ we still observe a logarithmic behavior the scaling with $\gamma$ but for $\gamma > \gamma_c$ the scaling with $L_A$ does not lead to data collapse. However, the data for $\gamma > \gamma_c$ as a function of $L$ tend to zero for large $\gamma$ and show that in this case $L_A\Delta S$ appears to go to zero (Fig.7(b)) in the thermodynamic limit, consistently with a bounded entanglement phase. This is clearly seen also in Fig.8, where the incremental ratio of the bipartite entanglement entropy as a function of the parameter $d$ is considered, for a fixed value of $\gamma > \gamma_c(d,h)$ for all $d$ here considered (see also Fig.9 for the behaviour of $c$). Notice that also in this case we observed that the imaginary part of the spectrum is gapless in correspondence of the logarithmic scaling of the entanglement entropy, gapfull otherwise.

## 5 Conclusions

We have analyzed measurement induced phase transitions in the zero-click trajectories for two different models, the Transverse Field Ising Model and the Long Range Kitaev chain in 1D. Starting from the Transverse Field Ising Model, we observed that the entanglement properties of the vacuum of the non-Hermitian Bogoliubov quasiparticles are the same of the steady state of the zero-click trajectories: the bipartite entanglement entropy of the vacuum state scales

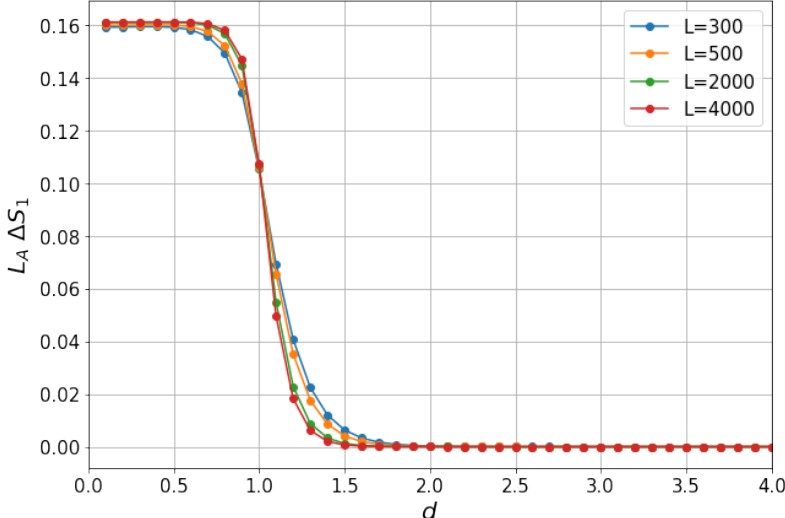

Figure 8: Incremental ratio of the bipartite entanglement entropy, scaled by the relative dimension of the intervals, as a function of $d$, for fixed value of $\gamma = 5$, $h = 0.1$. Here $\gamma > \gamma_c$ for every value of the parameter $d$ considered. Different dimensions of the global system were considered, while the partition is always $L_A = \frac{L}{5}$.

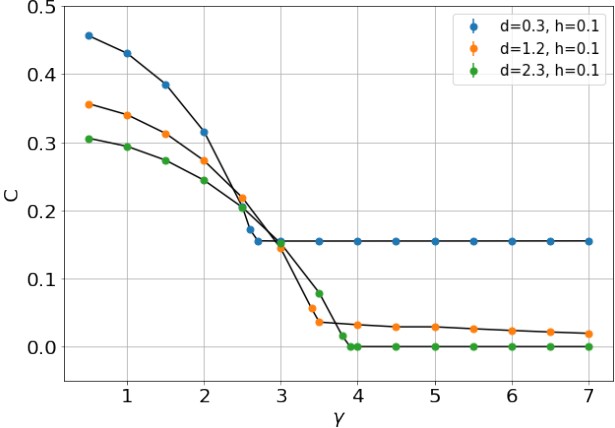

Figure 9: Variation of the parameter $c$ as a function of $\gamma$ for different values of $d$, where $S_1 = c \ln(L_A) + b$, obtained by means of fit of the scaling of the entanglement entropy as a function of $L_A$ in the vacuum.

logarithmically with the dimension of the subsystem if $\gamma < \gamma_c$, while if $\gamma > \gamma_c$ the bipartite entanglement entropy obeys an area law. The non-Hermitian non-decaying quasiparticles $\hat{\gamma}_q^*$ and $\hat{\gamma}_{-q}^*$ do not change the qualitative behaviour of the state in $|\psi\rangle = A|0_\gamma\rangle + B\hat{\gamma}_{-q}^* \hat{\gamma}_q^* |0_\gamma\rangle$. This motivated us to formulate a generalized area law conjecture: the entanglement entropy is bounded whenever the imaginary part of the spectrum of non-hermitean quasi-particles is gapfull, while it scales logarithmically with subsystem size when it is gapless. We prove that this is the case also in the perturbed no-click limit, a finite size approach that describes the steady state in the presence of rare jumps. Finally, we studied the entanglement phase transitions in the no-click limit of the Long-Range Kitaev model as a function of the power-law $d$ controlling the decay of the interaction with distance, $\sim \frac{1}{r^d}$. We observe that for $d < 1$ there is a transition between two logarithmic phases while for $d > 1$ the transition is between a logarithmic phase and an area law one. Also in this case the generalized area law conjecture appears to be satis-

fied. This results, together with those recently reported in Ref. [52] which further support the *generalized area law conjecture,* suggest that this connection could be framed in a more precise and mathematical language. We leave this issue to further investigations.

## Acknowledgements

We acknowledge discussions with R. Fazio, A.Paviglianiti, M.Schiró and X. Turkeshi.

## A    Diagonalization of the non-Hermitian Hamiltonian

In the no-click limit the effective Hamiltonian determining up to a c-number the evolution of the wave function for the Quantum Ising model is

$$H_{\text{QI}} = -J \sum_{i=1}^{N} \hat{\sigma}_i^x \hat{\sigma}_{i+1}^x - \left( h + i\frac{\gamma}{4} \right) \sum_{i}^{N} \hat{\sigma}_i^z, \tag{A.1}$$

for the Transverse Field Ising Model.

In the following we will diagonalize the non-Hermitian Hamiltonian by means of a Jordan-Wigner transformation followed by a Bogoliubov rotation.
Introducing the fermionic operators $c_i, c_i^\dagger$ through the relations

$$\hat{\sigma}_i^z = 1 - 2\hat{c}_i^\dagger \hat{c}_i, \qquad \hat{\sigma}_i^y = i\hat{K}_i(\hat{c}_i^\dagger - \hat{c}_i), \qquad \hat{\sigma}_i^x = \hat{K}_i(\hat{c}_i + \hat{c}_i^\dagger),$$
$$\hat{K}_i = \prod_{j<i}(1 - 2\hat{c}_j^\dagger \hat{c}_j), \tag{A.2}$$

we obtain

$$\hat{H} = -J \sum_i \hat{c}_i^\dagger \hat{c}_{i+1} + \hat{c}_i^\dagger \hat{c}_{i+1}^\dagger + h.c. - \left( h + i\frac{\gamma}{4} \right) \sum_i (1 - 2\hat{c}_i^\dagger \hat{c}_i), \tag{A.3}$$

which is integrable.

The diagonalization proceeds by first introducing the operators $\hat{c}_k$ and $\hat{c}_k^\dagger$ by means of the Fourier transform

$$\hat{c}_k = \frac{1}{\sqrt{L}} \sum_{R_i} e^{-ikR_i} \hat{c}_i, \qquad \hat{c}_i = \frac{1}{\sqrt{L}} \sum_k e^{ikR_i} \hat{c}_k. \tag{A.4}$$

Where the momenta depends on the parity of the system: we will use

$$\kappa_{PBC} = \left\{ k | k = \frac{2\pi n}{L}, n \in \left\{ -\frac{L}{2}, ...., \frac{L}{2} - 1 \right\} \right\}, \tag{A.5}$$

for the periodic boundary conditions (PBC), while

$$\kappa_{ABC} = \left\{ k | k = \frac{\pi(2n+1)}{L}, n \in \left\{ -\frac{L}{2}, ...., \frac{L}{2} - 1 \right\} \right\}, \tag{A.6}$$

for the antiperiodic boundary conditions (ABC). Without loss of generality we restrict ourselves to an initial state characterized by an even number of fermions; because of this, in the following analysis we will use ABC. The Fourier transform allows us to write the Hamiltonians $H_{QI}$ as

$$\hat{H} = \sum_{k>0} \hat{H}(k), \tag{A.7}$$

where

$$\hat{H}_{QI}(k) = \begin{pmatrix} \hat{c}_k^\dagger & \hat{c}_{-k} \end{pmatrix} \begin{pmatrix} 2(h - J\cos(k)) + i\frac{\gamma}{2} & 2iJ\sin(k) \\ -2iJ\sin(k) & -2(h - J\cos(k)) - i\frac{\gamma}{2} \end{pmatrix} \begin{pmatrix} \hat{c}_k \\ \hat{c}_{-k}^\dagger \end{pmatrix}. \tag{A.8}$$

The diagonalization of the hamiltonian proceeds now by rotating both $\hat{H}(k)$ in diagonal form and correspondigly introduce, through a generalized Bogoliubov transformation, the non-hermitian quasi-particles. For $H_{QI}$, first of all introduce the matrix $V_k$ diagonalizing $\hat{H}_k$, i.e. such as

$$V_k^{-1} \hat{H}_k V_k = \begin{pmatrix} \lambda_k & 0 \\ 0 & -\lambda_k \end{pmatrix}, \tag{A.9}$$

where the eigenvalue is expressed in terms of $a = 2(h - J\cos(k)) + i\frac{\gamma}{2}$, $b = 2iJ\sin(k)$ as

$$\lambda_k = \pm\sqrt{a^2 + |b|^2} = \pm\sqrt{4\left(h - J\cos k + i\frac{\gamma}{4}\right)^2 + 4\sin^2 k}, \tag{A.10}$$

the sign chosen in such a way that the sign of the imaginary part of $\lambda_k$ is negative. The matrix $V_k$ is in general non unitary [44] and is

$$V_k = \begin{pmatrix} u & -\frac{\lambda_k - a}{b^*} u^* \\ \frac{\lambda_k - a}{b} u & u^* \end{pmatrix}, \tag{A.11}$$

where $u = 1/\sqrt{1 + |\frac{\lambda_k - a}{b}|^2}$.

For the quantum Ising chain the non-hermitian quasi-particles are then [49, 50, 53]

$$\hat{\gamma}_k = \frac{1}{\det(V_k)}\left(u^* \hat{c}_k + \frac{\lambda_k - a}{b^*} u^* \hat{c}_{-k}^\dagger\right), \qquad \hat{\gamma}_{-k}^* = \frac{1}{\det(V_k)}\left(-\frac{\lambda_k - a}{b} u \hat{c}_k + u \hat{c}_{-k}^\dagger\right), \tag{A.12}$$

$$\hat{\gamma}_k^* = \left(u \hat{c}_k^\dagger + \frac{\lambda_k - a}{b} u \hat{c}_{-k}\right), \qquad \hat{\gamma}_{-k} = \left(-\frac{\lambda_k - a}{b^*} u^* \hat{c}_k^\dagger + u^* \hat{c}_{-k}\right). \tag{A.13}$$

The quasi-particle operators satisfy simple commutation relations

$$\{\hat{\gamma}_k, \hat{\gamma}_{-k}^*\} = 0, \qquad \{\hat{\gamma}_k, \hat{\gamma}_k^*\} = 1, \tag{A.14}$$

and diagonalize the Hamiltonian

$$\hat{H} = \sum_{k>0} \lambda_k \hat{\gamma}_k^* \hat{\gamma}_k - \lambda_k \hat{\gamma}_{-k} \hat{\gamma}_{-k}^* = \sum_{k>0} \lambda_k (\hat{\gamma}_k^* \hat{\gamma}_k + \hat{\gamma}_{-k}^* \hat{\gamma}_{-k}) - \Lambda_0, \tag{A.15}$$

with $\Lambda_0 = \sum_{k>0} \lambda_k$. Notice in particular that, by construction, the right vacuum defined as

$$\hat{\gamma}_k |0_\gamma\rangle = 0, \qquad \hat{\gamma}_{-k} |0_\gamma\rangle = 0 \tag{A.16}$$

is the state with largest imaginary part (not lowest real part). By construction, because of the normalization factor in Eq.(4), the stationary states of the dynamics will be therefore a linear combination of the vacuum and of quasi-particle states such that $\Gamma_k \equiv \text{Im}[\lambda_k] = 0$, while all amplitudes of the other quasi-particle states will decay to zero at long times.

An additional peculiarity related to the non-hermiticity of the operators is that

$$\langle 0_\gamma | \hat{\gamma}_k^* \neq 0, \qquad \langle 0_\gamma | \hat{\gamma}_{-k}^* \neq 0. \tag{A.17}$$

It is evident that an analogous construction can be made for the long range Kitaev Model,

$$H_{\mathrm{K}} = -J\sum_i (\hat{c}_i^\dagger \hat{c}_{i+1} + h.c) - \frac{J}{2}\sum_i \sum_{r=1}^{L-1}\frac{1}{l^d}\left[\hat{c}_i^\dagger \hat{c}_{i+r}^\dagger + h.c.\right] - \left(h + i\frac{\gamma}{4}\right)\sum_i(1 - 2\hat{c}_i^\dagger \hat{c}_i). \quad \text{(A.18)}$$

After the Fourier transform we obtain

$$\hat{H}_K = \sum_{k>0}\hat{H}_K(k), \quad \text{(A.19)}$$

where

$$H_K(k) = \begin{pmatrix}\hat{c}_k^\dagger & \hat{c}_{-k}\end{pmatrix}\begin{pmatrix}2(h - J\cos(k) + i\frac{\gamma}{2} & iJg_d(k) \\ -iJg_d(k) & -2(h - J\cos(k)) - i\frac{\gamma}{2}\end{pmatrix}\begin{pmatrix}\hat{c}_k \\ \hat{c}_{-k}^\dagger\end{pmatrix}, \quad \text{(A.20)}$$

and we have defined

$$g_d(k) = \sum_{r=1}^{L-1}\frac{\sin(kr)}{l^d}. \quad \text{(A.21)}$$

Also in this case a summation over the ABC set of momenta has been considered. Then a construction analogous to the one of the Transverse Field Ising Model can be made for $\hat{H}_K$ redefining

$$\begin{aligned} a &= 2(h - J\cos(k)) + i\frac{\gamma}{2}, \\ b &= iJg_d(k). \end{aligned} \quad \text{(A.22)}$$

# B The Majorana's Fermions correlation matrix

The Majorana's fermions correlation matrix is defined as

$$M_{mn} = \frac{\langle \breve{c}_m \breve{c}_n \rangle_\psi}{\langle \psi | \psi \rangle}, \quad \text{(B.1)}$$

where

$$\breve{c}_{2l-1} = \hat{c}_l^\dagger + \hat{c}_l, \qquad \breve{c}_{2l} = -i(\hat{c}_l^\dagger + \hat{c}_l).$$

In a gaussian state if we know the element of the Majorana fermions correlation matrix we can compute the value of every observables [54]; among them, the bipartite entanglement entropy, which is relevant in the analysis here reported. If the state that describes the system is $|\psi\rangle = A|0_\gamma\rangle + B\hat{\gamma}_{-q}^*\hat{\gamma}_q^*|0_\gamma\rangle$, A, B $\in$ **C**, and $N_k = |u|^2 + |\frac{\lambda_k - a}{b}u|^2$, then

$$\begin{aligned} \frac{\langle \breve{c}_{2m-1}\breve{c}_{2p-1}\rangle_\psi}{\langle \psi | \psi \rangle} &= \delta_{m,p} + \sum_{k>0}\frac{2i\sin k(R_p - R_m)}{L}\frac{2i\,\mathrm{Im}(\lambda_k - a)u^2}{b|N_k|^2} \\ &\quad + \frac{2i\sin q(R_m - R_p)}{L\langle \psi | \psi \rangle}\left[A^*B + B^*A\right] \\ &\quad + \frac{2i\sin q(R_p - R_m)}{L\langle \psi | \psi \rangle}\left(\frac{-2i\,\mathrm{Im}(\lambda_q - a)u^2}{b|N_q|^2}\right)\left(A^*B + B^*A\right), \quad \text{(B.2)} \end{aligned}$$

$$\frac{\langle \check{c}_{2m}\check{c}_{2p}\rangle_\psi}{\langle \psi|\psi\rangle} = \delta_{m,p} - \sum_{k>0} \frac{2i \sin k(R_p - R_m)}{L} \frac{2i \,\mathrm{Im}(\lambda_q - a)u^2}{b|N_q|^2}$$
$$+ \frac{2i \sin q(R_p - R_m)}{L\,\langle \psi|\psi\rangle}\big[A^*B + B^*A\big]$$
$$- \frac{2i \sin q(R_p - R_m)}{L\,\langle \psi|\psi\rangle}\left(\frac{-2i \,\mathrm{Im}(\lambda_q - a)u^2}{b|N_q|^2}\right)(A^*B + B^*A), \qquad \text{(B.3)}$$

$$\frac{\langle \check{c}_{2m-1}\check{c}_{2p}\rangle_\psi}{\langle \psi|\psi\rangle} = \frac{-i}{L}\sum_{k>0}\left[2\cos k(R_m - R_p)\frac{u^2 - |v|^2}{|N_k|^2} + 2i \sin k(R_m - R_p)\frac{2\,\mathrm{Re}(\lambda_k - a)u^2}{b|N_k|^2}\right]$$
$$+ i\frac{2}{L\det(V_q)\,\langle \psi|\psi\rangle}\left[2\cos q(R_m - R_p)\Big(u^2 + v^2\Big) + 4i \sin q(R_m - R_p)uv\right]$$
$$\times\left(|B|^2 + A^*B\frac{-2i \,\mathrm{Im}(\lambda_q - a)u^2}{b|N_q|^2}\right)$$
$$- \frac{i}{L\det(V_q)\,\langle \psi|\psi\rangle}\left[4\cos q(R_m - R_p)uv + 2i \sin q(R_p - R_m)\Big(u^2 + v^2\Big)\right]$$
$$\times\left[B^*A - A^*B + \left(\frac{4i \,\mathrm{Im}(\lambda_q - a)u^2}{b|N_q|^2}\right)|B|^2 - \left(\frac{2i \,\mathrm{Im}(\lambda_q - a)}{b|N_q|^2}u^2\right)^2(A^*B + B^*A)\right], \quad \text{(B.4)}$$

$$\frac{\langle \check{c}_{2m-1}\check{c}_{2p}\rangle_\psi}{\langle \psi|\psi\rangle} = \frac{i}{L}\sum_{k>0}\left[2\cos k(R_m - R_p)\frac{u^2 - |v|^2}{|N_k|^2} + 2i \sin k(R_p - R_m)\frac{2\,\mathrm{Re}(\lambda_k - a)u^2}{b|N_k|^2}\right]$$
$$- i\frac{2}{L\det(V_q)\,\langle \psi|\psi\rangle}\left[2\cos q(R_p - R_m)\Big(u^2 + v^2\Big) + 4i \sin q(R_p - R_m)uv\right]$$
$$\times\left(|B|^2 + A^*B\frac{-2i \,\mathrm{Im}(\lambda_q - a)u^2}{b|N_q|^2}\right)$$
$$+ \frac{i}{L\det(V_q)\,\langle \psi|\psi\rangle}\left[4\cos q(R_p - R_m)uv + 2i \sin q(R_m - R_p)\Big(u^2 + v^2\Big)\right]$$
$$\times\left[B^*A - A^*B + \left(\frac{4i \,\mathrm{Im}(\lambda_q - a)u^2}{b|N_q|^2}\right)|B|^2 - \left(\frac{2i \,\mathrm{Im}(\lambda_q - a)}{b|N_q|^2}u^2\right)^2(A^*B + B^*A)\right]. \quad \text{(B.5)}$$

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
