# Peer review of "Measurement phase transitions in the no-click limit as quantum phase transitions of a non-hermitean vacuum"

_SciPost Physics Core, doi:SciPost Phys. Core 6, 051 (2023)_

## Round 1 · Referee Report · Anonymous (Referee 1) · 2023-3-27

Strengths

  1. Suggests an easy and accessible way to study measurement-induced phase transitions, and proposes and extension of the area-law theorem to non-Hermitian Hamiltonians.
  2. Extensively discusses the validity of the assumptions underlying the work and provides numerical evidences to support the discussion.
  3. Provides sufficiently detailed analytics in order to reproduce the analysis and the results.
  4. Gives proper background and references to motivate this work

Weaknesses

  1. The manuscript suffers from some imprecisions on the numerical results
  2. The figures are not sufficiently described and commented in the main text

Report

In their manuscript Zerba and Silva provide a novel way to study measurement-induced phase transitions by the means of the no-click limit. In this regime, the dynamics of a system boils down to a non-unitary dynamics piloted by a non-Hermitian Hamiltonian. The authors show that the properties of the system are essentially captured by the non-Hermitian vaccum of the Hamiltonian. They also claim that the scaling of the bi-partite entanglement entropy with the subsystem size undergoes a phase transition that is correlated to the gapped or not nature of the spectrum of the Hamiltonian.

In order to support their claim, the authors provide numerical evidences based on the study of two one-dimensional integrable models : the transverse-field Ising model and the long-range Kitaev chain. They also test the consistency of their approach by introducing sparse clicks from the measurement device to which the many-body system is coupled to.

Despite the high interest of this work in the domain of measurement-induced phase transitions, the manuscript lacks clarity and an effort is required in order to properly highlight the novelty of these results.

Requested changes

  1. Most of the analytical results provided in Sec 2. could be moved in the Appendix for the sake of clarity.
  2. Figure 3 does enable the reader to 4assess whether the difference between $S_{zc}$ and $S_v$ is sizeable or not. I would instead suggest to plot the relative error $(S_{zc}-S_v)/S_{vc}$.
  3. How many trajectories where used to evaluate the average showed on Fig 5(c) ? It would also be useful to show on the same plot the $c$ as a function of $\gamma$ in the no-click case for comparison of the critical behaviour of the transition (position of the critical point, finite-size effects)

---

## Round 2 · Referee Report · Anonymous (Referee 1) · 2023-5-10

Report

In view of the modifications brought to this manuscript by its authors, which greatly improve the clarity of the text and further stress the relevance for the study of measurement-induced phase transitions, I would recommend the publication of this article in SciPost Physics Core.

---

## Round 2 · Author Response

Dear Editor,

We thank the referee for the report, for the appreciation of our work and for helping us with concrete suggestions to increase the readability of our paper. We have made revisions according to the referee suggestions, as reported in the "List of changes".

We hope that with these changes the paper will be accepted for publication in Scipost.

Sincerely,
Caterina Zerba, Alessandro Silva

---

## Round 2 · List of Changes

\\1. We have moved the analytic results in Sec. 2 to the Appendix. We kept in Sec 2 only the strict necessary to make the rest of the manuscript clear to the reader without necessarily going through the appendices. In particular in the revised version we write: "All the details concerning the diagonalization can be found in Appendix A . For both models the diagonalized Hamiltonian takes the form
\begin{equation}
\hat{H}=\sum_{k>0} \lambda_k \hat{\gamma}^*_k\hat{\gamma}_k - \lambda_k \hat{\gamma}_{-k}\hat{\gamma}_{-k}^*=\sum_{k>0} \lambda_k (\hat{\gamma}^*_k\hat{\gamma}_k + \hat{\gamma}_{-k}^*\hat{\gamma}_{-k})-\Lambda_0,\end{equation}
where $\Lambda_0=\sum_{k>0}\lambda_k$, $\hat{\gamma}$ are the non-hermitian quasiparticle annihilation operators and $\lambda_k$ are the ( complex ) eigenvalues which are specific of the model considered. We find that for the Quantum Ising model
\begin{equation}
\lambda_k= \pm \sqrt{4\bigg(h-J\cos{k}+i\frac{\gamma}{4}\bigg)^2+ 4 J^2\sin^2{k}},
\end{equation}
while for the Long Range Kitaev model
\begin{equation}
\lambda_k= \pm \sqrt{4\bigg(h-J\cos{k}+i\frac{\gamma}{4}\bigg)^2+ J^2 g_d(k)^2},
\end{equation}
where $g_d(k)=\sum_{r=1}^{L-1} \frac{\sin(k r)}{l^d}$. The sign is chosen in such a way that the sign of the imaginary part of $\lambda_k$ is negative [49]. Thus the non-hermitian Hamiltonian right vacuum
\begin{align}
\hat{\gamma}_k |0_\gamma\big>=0,\hspace{1 cm}
\hat{\gamma}_{-k}|0_\gamma\big>=0,
\end{align}
can always be construct as the state with largest imaginary part (not lowest real part). Because of the normalization factor in Eq.(4), the stationary states of the dynamics will be a linear combination of the vacuum and of quasi-particle states such that $\Gamma_k \equiv \Im[\lambda_k]=0$, while all amplitudes of the other quasi-particle states will decay to zero at long times.". In the Appendix the diagonalization of the Hamiltonian as reported in section 2 of the previous version is presented.\\
\\ 2. Figure 3 has been changed as requested: the quantity reported in the graph is the relative difference $(S_{zc}-S_{v})/S_{v}$. \\
\\3. In figure 5(c) we clarified how the parameter $c$ was computationally obtained by specifying in the caption of the figure the number of trajectories considered: "$N_{tr}=20$". In figure 5(c) we inserted a comparison with the result from the zero click trajectories and the vacuum. We commented on this comparison in the text by highlighting that "the transition point is however unchanged by the presence of rare jumps".\\

---

## Editorial Decision

published